# LEARNING PROTEIN SEQUENCE EMBEDDINGS USING INFORMATION FROM STRUCTURE

**Tristan Bepler**
Computational and Systems Biology
Computer Science and Artificial Intelligence Laboratory
Massachusetts Institute of Technology
Cambridge, MA 02139, USA
tbepler@mit.edu

**Bonnie Berger**
Computer Science and Artificial Intelligence Laboratory
Department of Mathematics
Massachusetts Institute of Technology
Cambridge, MA 02139, USA
bab@mit.edu

## ABSTRACT

Inferring the structural properties of a protein from its amino acid sequence is a challenging yet important problem in biology. Structures are not known for the vast majority of protein sequences, but structure is critical for understanding function. Existing approaches for detecting structural similarity between proteins from sequence are unable to recognize and exploit structural patterns when sequences have diverged too far, limiting our ability to transfer knowledge between structurally related proteins. We newly approach this problem through the lens of representation learning. We introduce a framework that maps any protein sequence to a sequence of vector embeddings — one per amino acid position — that encode structural information. We train bidirectional long short-term memory (LSTM) models on protein sequences with a two-part feedback mechanism that incorporates information from (i) global structural similarity between proteins and (ii) pairwise residue contact maps for individual proteins. To enable learning from structural similarity information, we define a novel similarity measure between arbitrary-length sequences of vector embeddings based on a soft symmetric alignment (SSA) between them. Our method is able to learn useful position-specific embeddings despite lacking direct observations of position-level correspondence between sequences. We show empirically that our multi-task framework outperforms other sequence-based methods and even a top-performing structure-based alignment method when predicting structural similarity, our goal. Finally, we demonstrate that our learned embeddings can be transferred to other protein sequence problems, improving the state-of-the-art in transmembrane domain prediction. [1]

## 1 INTRODUCTION

Proteins are linear chains of amino acid residues that fold into specific 3D conformations as a result of the physical properties of the amino acid sequence. These structures, in turn, determine the wide array of protein functions, from binding specificity to catalytic activity to localization within the cell. Information about structure is vital for studying the mechanisms of these molecular machines in health and disease, and for development of new therapeutics. However, experimental structure determination is costly and atomic structures have only been determined for a tiny fraction of known proteins. Methods for finding proteins with related structure directly from sequence are of considerable interest,

---

[1]source code and datasets are available at https://github.com/tbepler/protein-sequence-embedding-iclr2019

but the problem is challenging, because sequence similarity and structural similarity are only loosely related [1, 2, 3, 4], e.g. similar structural folds can be formed by diverse sequences. As a result, our ability to transfer knowledge between proteins with similar structures is limited.

In this work, we address this problem by learning protein sequence embeddings using weak supervision from global structural similarity for the first time. Specifically, we aim to learn a bidirectional LSTM (biLSTM) embedding model, mapping sequences of amino acids to sequences of vector representations, such that residues occurring in similar structural contexts will be close in embedding space. This is difficult, because we have not observed position-level correspondences between sequences, only global sequence similarity. We solve this by defining a whole sequence similarity measure from sequences of vector embeddings. The measure decomposes into an alignment of the sequences and pairwise comparison of the aligned positions in embedding space. For the alignment, we propose a soft symmetric alignment (SSA) mechanism — a symmetrization of the directional alignment commonly used in attention mechanisms. Furthermore, in order to take advantage of information about local structural context within proteins, we extend this framework to include position-level supervision from contacts between residues in the individual protein structures. This multitask framework (Figure 1) allows us to newly leverage both global structural similarity between proteins and residue-residue contacts within proteins for training embedding models.

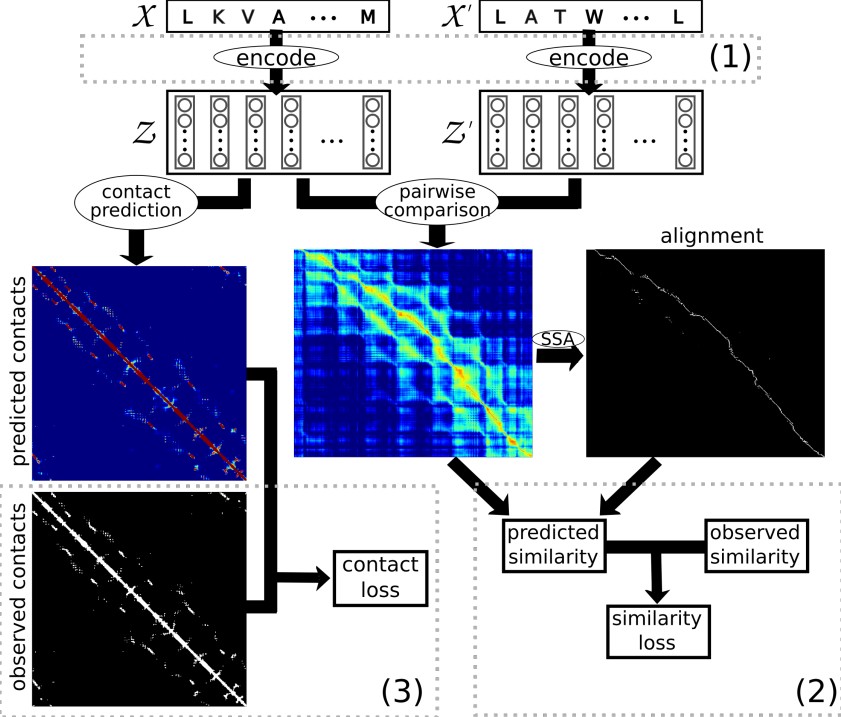

Figure 1: Diagram of the learning framework. (1) Amino acid sequences are transformed into sequences of vector embeddings by the encoder model. (2) The similarity prediction module takes pairs of proteins represented by their sequences of vector embeddings and predicts their shared SCOP level. Sequences are first aligned based on L1 distance between their vector embeddings using SSA. From the alignment, a similarity score is calculated and related to shared SCOP levels by ordinal regression. (3) The contact prediction module uses the sequence of vector embeddings to predict contacts between amino acid positions within each protein. The contact loss is calculated by comparing these predictions with contacts observed in the 3D structure of the protein. Error signal from both tasks is used to fit the parameters of the encoder.

We first benchmark our model's ability to correctly predict structural similarity between pairs of sequences using the SCOPe ASTRAL dataset [5]. This dataset contains protein domains manually classified into a hierarchy of structural categories (Appendix Figure 3). We show that our model dramatically outperforms other sequence-based protein comparison methods when predicting co-membership in the SCOP hierarchy. Remarkably, our model even outperforms TMalign [6], which

requires structures as input and therefore structures must be known *a priori*. In contrast, our model uses only sequence as input. Next, we perform an ablation study to evaluate the importance of our modeling components for structural similarity prediction. We also consider an additional task, secondary structure prediction, to assess the model's ability to capture local structure features. We demonstrate that SSA outperforms alternative alignment methods for both of these tasks and that inclusion of the contact prediction training task further improves performance.

Finally, we demonstrate that the embeddings learned by our model are generally applicable to other protein machine learning problems by leveraging our embeddings to improve the state-of-the-art in transmembrane prediction. This work presents the first attempt in learning protein sequence embeddings from structure and takes a step towards bridging the sequence-structure divide with representation learning.

## 2 RELATED WORK

Current work in protein sequence embeddings has primarily been focused on unsupervised k-mer co-occurence approaches to learn fixed sized vector representations [7, 8] based on similar methods in NLP [9, 10, 11, 12]. Melvin et al. [13] also learn fixed sized semantic embeddings by projecting alignment scores into a low dimensional vector to recapitulate rankings given by existing alignment tools and shared superfamily membership. However, fixed sized vector representations are limited, because they are not usable for any sequence labeling problems (e.g. active site prediction, transmembrane region prediction, etc.). Other methods have focused on manual feature engineering based on biophysical and sequence attributes [14]. These methods rely on expert knowledge and do not capture properties that emerge from interactions between amino acids.

Instead, we seek to learn embeddings that encode the full structural context in which each amino acid occurs. This is inspired partly by the recent success of unsupervised contextual embedding models using bidirectional recurrent neural network language models [15, 16] where word embeddings, learned as a function of their context, have been successfully transferred to other tasks. In particular, we apply a similar language model for the first time on protein sequences as part of our supervised framework. Supervised embedding models have also been trained for natural language inference (NLI) but produce only fixed sized embeddings [17, 18, 19].

At the same time, problems involving word alignment given matched sequences, such as cross lingual word embeddings and document similarity, have also been explored. Cross lingual word embeddings are learned from unaligned parallel text, where sentences are matched between languages but words are not. Kočiský et al. [20] learn bilingual word embeddings jointly with a FastAlign [21] word alignment model using expectation maximization. BilBOWA [22] learns cross lingual word embeddings using parallel sentences without word level alignments by assuming a uniform alignment between words. However, Gouws et al. [22] assume all pairings between words are equally likely and do not infer them from current values of the embeddings. Related methods have been developed for measuring similarity between documents based on their words. Word Mover's Distance (WMD) and its supervised variant align words between pairs of documents by solving an optimal transport problem given by distance between word vectors. However, these methods are not designed for learning neural network embedding models and the embeddings are not contextual. Furthermore, WMD alignments are prohibitively expensive when alignments must be computed at every optimization step, scaling as $O(p^3 \log p)$ where $p$ is the number of unique words.

Our SSA solves these problems via an alignment mechanism inspired by previous work using soft alignments and attention mechanisms for sequence modeling [23, 24]. Further elaborate directional alignments have been used for question answering and reading comprehension models [25, 26, 27] and for natural language inference [28, 29, 29]. Unlike these methods, however, our SSA method is both symmetric and memoryless. Furthermore, it is designed for learning interpretable embeddings based on a similarity measure between individual sequence elements. It is also fast and memory efficient - scaling with the product of the sequence lengths.

Protein fold recognition is the problem of classifying proteins into folds (for example, as defined by the SCOP database) based on their sequences. Approaches to this problem have largely been based on sequence homology using sequence similarity to classify structures based on close sequence matches [5, 30]. These methods are either direct sequence alignment tools [31, 32] or based on profile HMMs

in which multiple sequence alignments are first built by iterative search against a large sequence database, the multiple sequence alignments are converted into profile HMMs, and then sequences are compared using HMM-sequence or HMM-HMM alignments [33, 34]. However, these methods are only appropriate for matching proteins with high sequence similarity [2, 3, 30]. In contrast, we focus on learning protein sequence representations that directly capture structure information in an easily transferable manner. We hypothesize that this approach will improve our ability to detect structural similarity from sequence while also producing useful features for other learning tasks.

## 3 METHODS

In this section, we describe the three components of our framework (Figure 1) in detail: (1) the specific choice of embedding model, a multi-layer bidirectional LSTM with additional inputs from a pretrained LSTM language model, (2) soft symmetric alignment and ordinal regression components for relating sequences of vector representations for pairs of proteins to their global structural similarity, and (3) the pairwise feature vectors and convolutional neural network design for residue-residue contact prediction.

### 3.1 BiLSTM SEQUENCE ENCODER WITH PRETRAINED LANGUAGE MODEL

**BiLSTM encoder.**   The encoder takes a sequence of amino acids representing a protein and encodes it into a sequence of vector representations of the same length. To allow the vector representations at each position to be functions of all surrounding amino acids, we structure the encoder as a stack of bidirectional LSTMs followed by a linear layer projecting the outputs of the last biLSTM layer into the final embedding space (Appendix Figure 2).

**Pretrained language model.**   The concept of feeding LSTM language model representations as inputs for supervised learning problems as part of a larger neural network model has shown recent success in NLP but has not yet been tried for biological sequences. Inspired partially by the success of ELMo [16], we consider, in addition to 1-hot representations of the amino acids, the inclusion of the hidden layers of a pretrained bidirectional LSTM language model as inputs to the encoder described above. The language model is pretrained on the raw protein sequences in the protein families database (Pfam) [35] to predict the amino acid at each position of each protein given the previous amino acids and the following amino acids (see Appendix section A.1 for details).

Specifically, given the language model hidden states at each position, $i$, denoted as $h_i^{LM}$, and the 1-hot representation of the amino acid at those positions, $x_i$, we introduce a learned linear transformation of these representations with ReLU non-linearity,

$$h_i^{input} = \text{ReLU}(W^{LM}h_i^{LM} + W^x x_i + b),$$

which is passed as input to the biLSTM sequence encoder. The parameters $W^{LM}$, $W^x$, and $b$ are trained together with the parameters of the biLSTM encoder. The parameters of the language model itself are frozen during training. In experiments without the language model, $h_i^{LM}$ is set to zero for all positions.

### 3.2 PROTEIN STRUCTURE COMPARISON

The primary task we consider for training the sequence embedding model with structural information is the prediction of global structural similarity between protein sequences as defined by shared membership in the SCOP hierarchy. SCOP is an expertly curated database of protein domain structures in which protein domains are assigned into a hierarchy of structures (Appendix Figure 3). Specifically, we define this as a multiclass classification problem in which a pair of proteins is classified into no similarity, class level similarity, fold level similarity, superfamily level similarity, or family level similarity based on the most specific level of the SCOP hierarchy shared by those proteins. We encode these labels as $y \in \{0, 1, 2, 3, 4\}$ based on the number of levels shared (i.e. y=0 encodes no similarity, y=1 encodes class similarity, etc.). In the following two sections, we describe how protein sequences are compared based on their sequences of vector embeddings using soft symmetric alignment and then how this alignment score is used to predict the specific similarity class by taking advantage of the natural ordering of these classes in an ordinal regression framework.

### 3.2.1 SOFT SYMMETRIC ALIGNMENT

In order to calculate the similarity of two amino acid sequences given that each has been encoded into a sequence of vector representations, $z_1...z_n$ and $z'_1...z'_m$, we develop a soft symmetric alignment mechanism in which the similarity between two sequences is calculated based on their vector embeddings as

$$\hat{s} = -\frac{1}{A} \sum_{i=1}^{n} \sum_{j=1}^{m} a_{ij} ||z_i - z'_j||_1 \tag{1}$$

where $a_{ij}$ are entries of the alignment matrix given by

$$\alpha_{ij} = \frac{\exp(-||z_i - z'_j||_1)}{\sum_{k=1}^{n} \exp(-||z_i - z'_k||_1)}, \qquad \beta_{ij} = \frac{\exp(-||z_i - z'_j||_1)}{\sum_{k=1}^{m} \exp(-||z_k - z'_j||_1)}, \text{ and}$$

$$a_{ij} = \alpha_{ij} + \beta_{ij} - \alpha_{ij}\beta_{ij} \tag{2}$$

and $A = \sum_{i=1}^{n} \sum_{j=1}^{m} a_{ij}$ is the length of the alignment.

### 3.2.2 ORDINAL REGRESSION

Next, to relate this scalar similarity to the ordinal structural similarities defined using SCOP ($y \in \{0, 1, 2, 3, 4\}$), we adopt an ordinal regression framework. Specifically, we learn a series of binary classifiers to predict whether the structural similarity level is greater than or equal to each level, $t$, given the alignment score (Equation 1). Given parameters $\theta_1...\theta_4$ and $b_1...b_4$, the probability that two sequences share similarity greater than or equal to $t$ is defined by $\hat{p}(y \geq t) = \text{sigmoid}(\theta_t \hat{s} + b_t)$, with the constraint that $\theta_t \geq 0$ to enforce that $\hat{p}$ increases monotonically with $\hat{s}$. The structural similarity loss is then given by

$$L^{similarity} = \mathop{\mathbb{E}}_{x,x'} \left[ \sum_{t=1}^{4} (y \geq t)\log(\hat{p}(y \geq t)) + (y < t)\log(1 - \hat{p}(y \geq t)) \right]. \tag{3}$$

These parameters are fit jointly with the parameters of the sequence encoder by backpropagating through the SSA which is fully differentiable. Furthermore, given these classifiers, the predicted probability that two sequences belong to structural similarity level $t$ is $\hat{p}(y = t) = \hat{p}(y \geq t)(1-p(y \geq t+1))$ with $\hat{p}(y \geq 0) = 1$ by definition.

### 3.3 RESIDUE-RESIDUE CONTACT PREDICTION

We can augment our SSA framework, in which position-level correspondence is inferred between sequences, with position-level supervision directly in the form of within protein contacts between residues. We introduce a secondary task of within protein residue-residue contact prediction with the hypothesis that the fine-grained structural supervision provided by the observed contacts will improve the quality of the embeddings. Contact prediction is a binary classification problem in which we seek to predict whether residues at positions $i$ and $j$ within an amino acid sequence make contact in the 3D structure. Following common practice in the protein structure field, we define two positions as making contact if the $C\alpha$ atoms of those residues occur within 8Åin the 3D structure.

In order to predict contacts from the sequence of embedding vectors given by the encoder for an arbitrary protein of length N, we define a pairwise features tensor of size (NxNx2D) where D is the dimension of the embedding vector containing pairwise features given by the concatenation of the absolute element-wise differences and the element-wise products of the vector representations for each pair of positions, $v_{ij} = [|z_i - z_j|; z_i \odot z_j]$. We choose this featurization because it is symmetric, $v_{ij} = v_{ji}$, and has shown widespread utility for pairwise comparison models in NLP [36]. These vectors are then transformed through a single hidden layer of dimension H (implemented as a width 1 convolutional layer) and ReLU activation giving $h_{ij} = \text{ReLU}(Wv_{ij} + b)$. Contact predictions

are then made by convolving a single 7x7 filter over the resulting NxNxH tensor with padding and sigmoid activation to give an NxN matrix containing the predicted probability for each pair of residues forming a contact.

Given the observed contacts, we define the contact prediction loss, $L^{contact}$, to be the expectation of the cross entropy between the observed labels and the predicted contact probabilities taken over all pairs of residues within each protein in the dataset.

**Complete multitask loss.** We define the full multitask objective by

$$\lambda L^{similarity} + (1 - \lambda)L^{contact} \tag{4}$$

where $\lambda$ is a parameter that interpolates between the structural similarity and contact prediction losses. This error signal is backpropogated through the contact prediction specific parameters defined in section 3.3, the similarity prediction specific parameters defined in section 3.2, and the parameters of sequence encoder defined in section 3.1 to train the entire model end-to-end.

### 3.4 HYPERPARAMETERS AND TRAINING DETAILS

Our encoder consists of 3 biLSTM layers with 512 hidden units each and a final output embedding dimension of 100 (Appendix Figure 2). Language model hidden states are projected into a 512 dimension vector before being fed into the encoder. In the contact prediction module, we use a hidden layer with dimension 50. These hyperparameters were chosen to be as large as possible while fitting on a single GPU with reasonable minibatch size. While we compare performance with simpler encoder architectures in section 4.2, it is possible that performance could be improved further with careful architecture search. However, that is beyond the scope of this work.

Sequence embedding models are trained for 100 epochs using ADAM with a learning rate of 0.001 and otherwise default parameters provided by PyTorch. Each epoch consists of 100,000 examples sampled from the SCOP structural similarity training set with smoothing of the similarity level distribution of 0.5. In other words, the probability of sampling a pair of sequences with similarity level $t$ is proportional to $N_t^{0.5}$ where $N_t$ is the number of sequence pairs with $t$ similarity in the training set. This is to slightly upweight sampling of highly similar pairs of sequences that would otherwise be rare. We choose 0.5 specifically such that a minibatch of size 64 is expected to contain two pairs of sequences with family level similarity. The structural similarity component of the loss is estimated with minibatches of 64 pairs of sequences. When using the full multitask objective, the contact prediction component uses minibatches of 10 sequences and $\lambda = 0.1$. Furthermore, during training we apply a small perturbation to the sequences by resampling the amino acid at each position from the uniform distribution with probability 0.05. These hyperparameters were selected using a validaton set as described in Appendix section A.2. All models were implemented in PyTorch and trained on a single NVIDIA Tesla V100 GPU. Each model took roughly 3 days to train and required 16 GB of GPU RAM. Additional runtime and memory details can be found in Appendix A.3.

In the following sections, we refer to the 3-layer biLSTM encoder trained with the full framework as "SSA (full)" and the framework without contact prediction (i.e. $\lambda = 1$) as "SSA (no contact prediction)."

## 4 RESULTS

### 4.1 STRUCTURAL SIMILARITY PREDICTION ON THE SCOP DATABASE

We first evaluate the performance of our full SSA embedding model for predicting structural similarity between amino acid sequences using the SCOP dataset. We benchmark our embedding model against several widely used sequence-based protein comparison methods, Needleman-Wunsch alignment (NW-align), phmmer [33], an HMM-to-sequence comparison method, and HHalign [34], an HMM-to-HMM comparison method. For HHalign, profle HMMs for each sequence were constructed by iterative search against the uniref30 database. We also benchmark our method agains TMalign, a method for evaluating protein similarity based on alignments of protein structures. Details for each of these baselines can be found in Appendix section A.4. Methods are compared based on accuracy

| Model | Accuracy | Correlation | | Average precision score | | | |
| --- | --- | --- | --- | --- | --- | --- | --- |
| | | $r$ | $\rho$ | **Class** | **Fold** | **Superfamily** | **Family** |
| ASTRAL 2.06 test set | | | | | | | |
|     NW-align | 0.78462 | 0.18854 | 0.14046 | 0.30898 | 0.40875 | 0.58435 | 0.52703 |
|     phmmer [HMMER 3.2.1] | 0.78454 | 0.21657 | 0.06857 | 0.26022 | 0.34655 | 0.53576 | 0.50316 |
|     HHalign [HHsuite 3.0.0] | 0.78851 | 0.36759 | 0.23240 | 0.40347 | 0.62065 | 0.86444 | 0.52220 |
|     TMalign | 0.80831 | 0.61687 | 0.37405 | 0.54866 | 0.85072 | 0.83340 | 0.57059 |
|     SSA (full) | **0.95149** | **0.90954** | **0.69018** | **0.91458** | **0.90229** | **0.95262** | **0.64781** |
| ASTRAL 2.07 new test set | | | | | | | |
|     NW-align | 0.80842 | 0.37671 | 0.23101 | 0.43953 | 0.77081 | 0.86631 | 0.82442 |
|     phmmer [HMMER 3.2.1] | 0.80907 | 0.65326 | 0.25063 | 0.38253 | 0.72475 | 0.82879 | 0.81116 |
|     HHalign [HHsuite 3.0.0] | 0.80883 | 0.68831 | 0.27032 | 0.47761 | 0.83886 | 0.94122 | 0.82284 |
|     TMalign | 0.81275 | 0.81354 | 0.39702 | 0.59277 | 0.91588 | 0.93936 | 0.82301 |
|     SSA (full) | **0.93151** | **0.92900** | **0.66860** | **0.89444** | **0.93966** | **0.96266** | **0.86602** |

Table 1: Comparison of the full SSA model with three protein sequence alignment methods (NW-align, phmmer, and HHalign) and the structure alignment method TMalign. We measure accuracy, Pearson's correlation ($r$), Spearman's rank correlation ($\rho$), and average precision scores for retrieving protein pairs with structural similarity of at least class, fold, superfamily, and family levels.

of classifying the shared SCOP level, correlation between the similarity score and shared SCOP level, and average precision scores when considering correct matches at each level of the SCOP hierarchy (i.e. proteins that are in the same class, proteins in the same fold, etc.).

The SCOP benchmark datasets are formed by splitting the SCOPe ASTRAL 2.06 dataset, filtered to a maximum sequence identity of 95%, into 22,408 train and 5,602 heldout sequences. From the heldout sequences, we randomly sample 100,000 pairs as the ASTRAL 2.06 structural similarity test set. Furthermore, we define a second test set using the newest release of SCOPe (2.07) by collecting all protein sequences added between the 2.06 and 2.07 ASTRAL releases. This gives a set of 688 protein sequences all pairs of which define the ASTRAL 2.07 new test set. The average percent identity between pairs of sequences within all three datasets is 13%. Sequence length statistics can be found in Appendix Table 4.

We find that our full SSA embedding model outperforms all other methods on all metrics on both datasets. On the 2.06 test set, we improve overall prediction accuracy from 0.79 to 0.95, Pearson's correlation from 0.37 to 0.91, and Spearman's rank correlation from 0.23 to 0.69 over the next best sequence comparison method, HHalign, without requiring any database search to construct sequence profiles. Furthermore, our full SSA model is much better for retrieving proteins sharing the same fold — the structural level of most interest for finding distant protein homologues — improving the average precision score by 0.28 on the 2.06 test set and 0.10 on the 2.07 test set over HHalign. We find that our full SSA embedding model even outperforms TMalign, a method for comparing proteins based on their 3D structures, when predicting shared SCOP membership. This is remarkable considering that our model uses only sequence information when making predictions whereas TMalign is provided with the known protein structures. The largest improvement comes at the SCOP class level where TMalign achieves much lower average precision score for retrieving these weak structural matches.

## 4.2 Ablation study of framework components

We next evaluate the individual model components on two tasks: structure similarity prediction on the ASTRAL 2.06 test set and 8-class secondary structure prediction on a 40% sequence identity filtered dataset containing 22,086 protein sequences from the protein data bank (PDB) [37], a repository of experimentally determined protein structures. Secondary structure prediction is a sequence labeling problem in which we attempt to classify every position of a protein sequence into one of eight classes describing the local 3D structure at that residue. We use this task to measure the utility of our embeddings for position specific prediction problems. For this problem, we split the secondary

| Embedding Model/Features | Structural similarity | | | Secondary structure | |
|---|---|---|---|---|---|
| | **Accuracy** | $r$ | $\rho$ | **Perplexity** | **Accuracy** |
| 1-mer | - | - | - | 4.804 | 0.374 |
| 3-mer | - | - | - | 4.222 | 0.444 |
| 5-mer | - | - | - | 5.154 | 0.408 |
| SSA (no contact prediction, no LM) | 0.89847 | 0.81459 | 0.64693 | 3.818 | 0.511 |
| ME (no contact prediction) | 0.92821 | 0.85977 | 0.67122 | 4.058 | 0.480 |
| UA (no contact prediction) | 0.93524 | 0.87536 | 0.67017 | 4.470 | 0.427 |
| SSA (no contact prediction) | 0.93794 | 0.88048 | 0.67645 | 4.027 | 0.487 |
| SSA (full) | **0.95149** | **0.90954** | **0.69018** | **2.861** | **0.630** |

Table 2: Study of individual model components. Results of structural similarity prediction on the ASTRAL 2.06 test set and secondary structure prediction are provided for embedding models trained with various components of our multitask framework. The SSA model trained without the language model component of the encoder and without contact prediction (SSA (without language model)), the SSA, UA, and ME models trained without contact prediction (ME, UA, SSA (without contact prediction)), and the full SSA embedding model.

structure dataset into 15,461 training and 6,625 testing sequences. We then treat each position of each sequence as an independent datapoint with features either given by the 100-d embedding vector or 1-hot encoding of the k-mer at that position and train a fully connected neural network (2 hidden layers, 1024 units each, ReLU activations) to predict the secondary structure class from the feature vector. These models are trained with cross entropy loss for 10 epochs using ADAM with learning rate 0.001 and a minibatch size of 256.

**SSA outperforms alternative comparison methods.** We first demonstrate the importance of our SSA mechanism when training the contextual embedding model by comparing the performance of biLSTM encoders trained with SSA versus the same encoders trained with uniform alignment and a mean embedding comparison approaches [22]. In uniform alignment (UA), we consider a uniform prior over possible alignments giving the similarity score. For the mean embedding method (ME), we instead calculate the similarity score based on the difference between the average embedding of each sequence. For these baselines, we substitute

$$\hat{s}^{UA} = -\frac{1}{nm} \sum_{i=1}^{n} \sum_{j=1}^{m} ||z_i - z'_j||_1 \qquad \text{and} \qquad \hat{s}^{ME} = -\left\|\left| \frac{1}{n} \sum_{i=1}^{n} z_i - \frac{1}{m} \sum_{j=1}^{m} z'_j \right|\right\|_1$$

in for SSA similarity (Equation 1) respectively during model training and prediction. These models are trained without contact prediction ($\lambda = 1$) to compare the alignment component in isolation.

We find that not only are the SSA embeddings better predictors of secondary structure than k-mer features (accuracy 0.487 vs. 0.444 for 3-mers), but that the SSA mechanism is necessary for achieving best performance on both the structure similarity and local structure prediction tasks. As seen in table 2, the ME model achieves close to SSA performance on secondary structure prediction, but is significantly worse for SCOP similarity prediction. The UA model, on the other hand, is close to SSA on SCOP similarity but much worse when predicting secondary structure. This suggests that our SSA mechanism captures the best of both methods, allowing embeddings to be position specific as in the ME model but also being better predictors of SCOP similarity as in the UA model.

**Contact prediction improves embeddings.** Although the SSA mechanism allows our embedding model to capture position specific information, we wanted to explore whether positional information *within* sequences in the form of contact prediction could be used to improve the embeddings. We train models with and without the contact prediction task and find that including contact prediction improves both the structural similarity prediction and secondary structure prediction results. The accuracy of secondary structure prediction improves from 0.487 without to 0.630 with contact prediction (Table 2). This suggests that the contact prediction task dramatically improves the quality

| Method | Prediction category | | | | |
| | TM | SP+TM | Globular | Globular+SP | Overall |
|---|---|---|---|---|---|
| TOPCONS | **0.80** | 0.80 | 0.97 | 0.91 | 0.87 |
| MEMSAT-SVM | 0.67 | 0.52 | 0.88 | 0.00 | 0.52 |
| Philius | 0.70 | 0.75 | 0.94 | 0.94 | 0.83 |
| Phobius | 0.55 | 0.83 | 0.95 | 0.94 | 0.82 |
| PolyPhobius | 0.55 | 0.83 | 0.95 | 0.94 | 0.82 |
| SPOCTOPUS | 0.71 | 0.78 | 0.78 | 0.79 | 0.76 |
| biLSTM+CRF (1-hot) | 0.32 | 0.34 | 0.99 | 0.46 | 0.52 |
| biLSTM+CRF (SSA (full)) | 0.77 | **0.85** | **1.00** | 0.94 | **0.89** |

Table 3: Accuracy of transmembrane prediction using structural embeddings in 10-fold cross valida-
tion and comparison with other transmembrane prediction methods. BiLSTM+CRF models using
either our full SSA model embeddings or 1-hot encodings of the amino acids as features are displayed
below the dotted line. We compare with results for a variety of transmembrane prediction methods
previously reported on the TOPCONS dataset.

of the local embeddings on top of the weak supervision provided by whole structure comparison. For
reference, we also report contact prediction performance for our full SSA model in Appendix A.5.

**Encoder architecture and pretrained language model are important.** Finally, we show, for the
first time with biological sequences, that a language model pretrained on a large unsupervised protein
sequence database can be used to transfer information to supervised sequence modeling problems.
SCOP similarity classification results for SSA embedding models trained with and without LM
hidden layer inputs shows that including the LM substantially improves performance, increasing
accuracy from 0.898 to 0.938. Furthermore, we examine the extent to which LM hidden states capture
all useful structural information by training SSA embedding models with less expressive power than
our 3-layer biLSTM architecture (Appendix Table 5). We find that the LM hidden states are not
sufficient for high performance on the structural similarity task with linear, fully connected (i.e. width
1 convolution), and single layer biLSTM embedding models having lower accuracy, Pearson, and
Spearman correlations than the 3-layer biLSTM on the ASTRAL 2.06 test set.

## 4.3 TRANSMEMBRANE PREDICTION

We demonstrate the potential utility of our protein sequence embedding model for transfering
structural information to other sequence prediction problems by leveraging our embedding model
for transmembrane prediction. In transmembrane prediction, we wish to detect which, if any,
segments of the amino acid sequence cross the lipid bilayer for proteins integrated into the cell
membrane. This is a well studied problem in protein biology with methods generally consisting
of HMMs with sophisticated, manually designed hidden state transition distributions and emission
distributions including information about residue identity, amino acid frequencies from multiple
sequence alignments, local structure, and chemical properties. Newer methods are also interested
in detection of signal peptides, which are short amino acid stretches at the beginning of a protein
sequence signaling for this protein to be inserted into the cell membrane.

To benchmark our embedding vectors for this problem, we develop a conditional random field (CRF)
model in which the propensity of each hidden state given the sequence of embedding vectors is
defined by a single layer biLSTM with 150 units. As a baseline, we include an identical biLSTM +
CRF model using only 1-hot encodings of the amino acids as features. For the transition probabilities
between states, we adopt the same structure as used in TOPCONS [38] and perform 10-fold cross
validation on the TOPCONS transmembrane benchmark dataset. We report results for correctly
predicting regions in proteins with only transmembrane domains (TM), transmembrane domains and
a signal peptide (SP+TM), neither transmembrane nor signal peptide domains (Globular), or a signal
peptide but no transmembrane regions (Globular+SP). Transmembrane state labels are predicted
with Viterbi decoding. Again following TOPCONS, predictions are counted as correct if, for TM

proteins, our model predicts no signal peptide, the same number of transmembrane regions, and those regions overlap with real regions by at least five positions. Correct SP+TM predictions are defined in the same way except that proteins must be predicted to start with a signal peptide. Globular protein predictions are correct if no transmembrane or signal peptides are predicted and Globular+SP predictions are correct if only a leading signal peptide is predicted.

We find that our transmembrane predictions rank first or tied for first in 3 out of the 4 categories (SP+TM, Globular, and Globular+SP) and ranks second for the TM category. Overall, our transmembrane predictions are best with prediction accuracy of 0.89 vs 0.87 for TOPCONS. Remarkably, this is by simply replacing the potential function in the CRF with a function of our embedding vectors, the hidden state grammar is the same as that of TOPCONS. Furthermore, the performance cannot be attributed solely to the biLSTM+CRF structure, as the biLSTM+CRF with 1-hot encoding of the amino acids performs poorly, tying MEMSAT-SVM for worst performance. This is particularly noteworthy, because TOPCONS is a meta-predictor. It uses outputs from a wide variety of other transmembrane prediction methods to define the transmembrane state potentials.

## 5 CONCLUSION

In this work, we proposed a novel alignment approach to learning contextual sequence embeddings with weak supervision from a global similarity measure. Our SSA model is fully differentiable, fast to compute, and can be augmented with position-level structural information. It outperforms competition in predicting protein structural similarity including, remarkably, structure alignment with TMalign. One consideration of training using SCOP, however, is that we focus exclusively on single-domain protein sequences. This means that the highly contextual embeddings given by the biLSTM encoder to single domains may differ from embeddings for the same domain in a multi-domain sequence. One interesting extension would thus be to modify the encoder architecture or training procedure to better model domains in multi-domain contexts. Nonetheless, the resulting embeddings are widely useful, allowing us to improve over the state-of-the-art in transmembrane region prediction, and can easily be applied to other protein prediction tasks such as predicting functional properties, active site locations, protein-protein interactions, etc. Most methods that use HMM sequence profiles or position-specific scoring matrices could be augmented with our embeddings. The broader framework extends to other related (non-biological) tasks.

## 6 ACKNOWLEDGMENTS

We would like to thank Tommi Jaakkola, Rohit Singh, Perry Palmedo, and members of the Berger lab for their helpful feedback.

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

# A APPENDIX

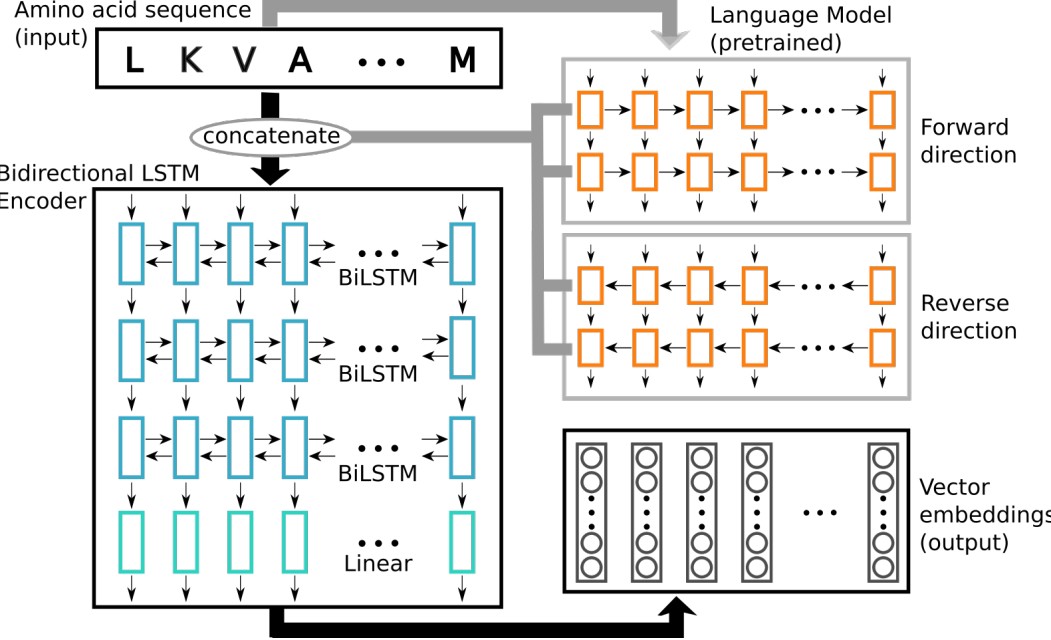

Figure 2: Illustration of the embedding model. The amino acid sequence is first passed through the pretrained language model in both forward and reverse directions. The hidden states at each position of both directions of the language model are concatenated together with a one hot representation of the amino acids and passed as input to the encoder. The final vector representations of each position of the amino acid sequence are given by a linear transformation of the outputs of the final bidirectional LSTM layer.

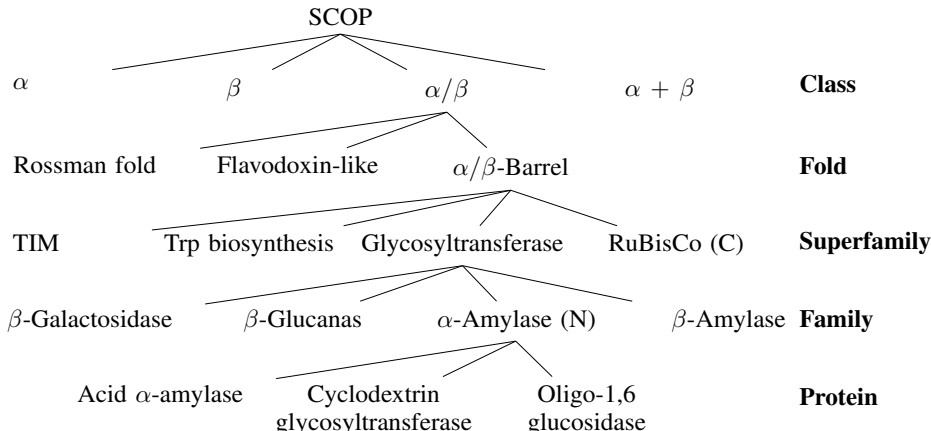

Figure 3: Illustration of the SCOP hierarchy modified from Hubbard et al. [39].

| Dataset | Sequence Length | | | |
| --- | --- | --- | --- | --- |
| | Mean | Std. Dev. | Min | Max |
| 2.06 train | 176 | 110 | 20 | 1,449 |
| 2.06 test | 180 | 114 | 21 | 1,500 |
| 2.07 new test | 190 | 149 | 25 | 1,664 |

Table 4: Sequence length statistics for the SCOPe datasets. We report the mean and standard deviation along with minimum and maximum sequence lengths.

| Embedding Model | Accuracy | $r$ | $\rho$ |
| --- | --- | --- | --- |
| Linear | 0.85277 | 0.74419 | 0.60333 |
| Fully connected (1-layer, 512 units) | 0.91013 | 0.84193 | 0.67024 |
| BiLSTM (1-layer) | 0.92964 | 0.87239 | 0.67485 |
| BiLSTM (3-layer) | **0.93794** | **0.88048** | **0.67645** |

Table 5: Comparison of encoder architectures on the ASTRAL 2.06 test set. Encoders included LM inputs and were trained using SSA without contact prediction.

## A.1 LANGUAGE MODEL TRAINING

The bidirectional LSTM language model was trained on the full set of protein domain sequences in the Pfam database, 21,827,419 total sequences. The language model was trained to predict the amino acid at position $i$ given observations of all amino acids before $i$ and all amino acids after $i$ by minimizing the cross entropy loss with log predicted log probabilities given by the sum of the forward and reverse LM direction predictions

$$\mathrm{log}p(x_i|x_{-i}) = \mathrm{log}p^F(x_i) + \mathrm{log}p^R(x_i)$$

where $p^F(x_i)$ is the probability given by the forward direction LSTM and $p^R(x_i)$ is the probability given by the reverse direction LSTM.

The language model architecture consisted of a 2-layer LSTM with 1024 units in each layer followed by a linear transformation into the 20-d amino acid prediction. All parameters were shared between the forward and reverse direction components. The model was trained for a single epoch using ADAM with a learning rate of 0.001 and minibatch size of 32.

## A.2 HYPERPARAMETER SELECTION

We select the resampling probability and $\lambda$ hyperparameters based on structural similarity prediction accuracy on a validation set held out from the SCOP ASTRAL 2.06 training set (Section 4.1). For these experiments, we hold out 2,240 random sequences from the 22,408 sequences of the training set. From these held out sequences, we randomly sample 100,000 pairs as the validation set.

| Encoder | Resampling | Contact Prediction | Accuracy |
|---|---|---|---|
| 3-layer biLSTM (no LM) | no | no | 0.89770 |
| 3-layer biLSTM (no LM) | yes | no | 0.90011 |
| 3-layer biLSTM | no | no | 0.92838 |
| 3-layer biLSTM | yes | no | 0.93375 |
| 3-layer biLSTM | yes | $\lambda = 0.5$ | 0.94181 |
| 3-layer biLSTM | yes | $\lambda = 0.33$ | 0.94474 |
| 3-layer biLSTM | yes | $\lambda = 0.1$ | 0.94749 |

Table 6: Evaluation of amino acid resampling probability and contact prediction loss weight, $\lambda$, for structural similarity prediction on the validation set. **(Top)** Resampling probability of 0.05 is compared with no resampling for 3-layer biLSTM encoders with and without language model components that are trained using SSA without contact prediction. **(Bottom)** Comparison of models trained using the full framework with $\lambda = 0.5$, $\lambda = 0.33$, and $\lambda = 0.1$.

**Resampling probability.** We consider models trained with amino acid resampling probability 0.05 and without amino acid resampling in the simplified framework (SSA without contact prediction) using the 3-layer biLSTM encoder with and without the language model component. We find that the structural similarity results are slightly improved when using amino acid resampling regardless of whether the LM component of the encoder is included (Appendix Table 6). Based on this result, all models were trained with amino acid resampling of 0.05.

**Multitask loss weight, $\lambda$.** We evaluate three values of $\lambda$, interpolating between the structural similarity loss and contact prediction loss, for predicting structural similarity on the validation set. Prediction accuracy increased progressively as $\lambda$ decreased and all models trained with contact prediction outperformed those trained without contact prediction (Appendix Table 6). Because it gave the best prediction accuracy on the validation set, $\lambda = 0.1$ was selected for training models with our full framework.

### A.3    TIME AND MEMORY REQUIREMENTS

Embedding time and memory scale linearly with sequence length. Embedding time is very fast ( 0.03 ms per amino acid on an NVIDIA V100) and easily fits on a single GPU (<9GB of RAM to embed 130,000 amino acids). Computing the SSA for sequence comparison scales as the product of the sequence lengths, O(nm), but is easily parallelized on a GPU. Computing the SSA for all 237,016 pairs of sequences in the SCOPe ASTRAL 2.07 new test set required on average 0.43 ms per pair (101 seconds total) when each comparison was calculated serially on a single NVIDIA V100. The contact prediction component scales quadratically with sequence length for both time and memory.

### A.4    STRUCTURAL SIMILARITY PREDICTION BENCHMARKS

For the NW-align method, similarity between protein sequences was computed using the BLOSUM62 substitution matrix with gap open and extend penalties of -11 and -1 respectively. For phmmer, each pair of sequences was compared in both directions (i.e. query->target and target->query) using the '–max' option. The similarity score for each pair was treated as the average off the query->target and target->query scores. For HHalign, multiple sequence alignments were first built for each sequence by using HHblits to search for similar sequences in the uniclust30 database with a maximum of 2 rounds of iteration (-n 2). Sequences pairs were then scored by using HHalign to score the target->query and query->target HMM-HMM alignments. Again, the average of the two scores was treated as the overall HHalign score. Finally, for TMalign, the structures for each pair of proteins were aligned and the scores for the query->target and target->query alignments were averaged to give the overall TMalign score for each pair of proteins.

To calculate the classification accuracy from the above scores, thresholds were found to maximize prediction accuracy when binning scores into similarity levels using 100,000 pairs of sequences sampled from the ASTRAL 2.06 training set.

### A.5    CONTACT PREDICTION PERFORMANCE

Although we use contact prediction as an auxiliary task to provide position-level structural supervision for the purpose of improving embedding quality and structure similarity prediction, we include results for predicting contacts using the trained contact prediction module here. We report results for contact prediction using the full SSA model on the SCOP ASTRAL 2.06 test set and 2.07 new test set in Appendix Table 7. Precision, recall, and F1 score are calculated using a probability threshold of 0.5 for assigning predicted contacts. We consider performance for predicting all contacts (i.e. contacts between all amino acid positions, excluding neighbors, $|i - j| \geq 2$), our training objective, and for distant contacts ($|i - j| \geq 12$) which are the focus of co-evolution based methods. We also report the precision of the top $L$, $L/2$, and $L/5$ contact predictions, where $L$ is the length of the protein sequence.

| Dataset | Contacts | Precision | Recall | F1 | AUPR | Pr @ L | Pr @ L/2 | Pr @ L/5 |
|---------|----------|-----------|--------|-----|------|--------|----------|----------|
| 2.06 test | All | 0.87299 | 0.72486 | 0.78379 | 0.84897 | 0.99659 | 0.99867 | 0.99911 |
| 2.06 test | Distant | 0.71814 | 0.47432 | 0.52521 | 0.60198 | 0.62536 | 0.72084 | 0.78983 |
| 2.07 new test | All | 0.87907 | 0.72888 | 0.78874 | 0.84577 | 0.99471 | 0.99879 | 0.99914 |
| 2.07 new test | Distant | 0.70281 | 0.46570 | 0.50454 | 0.57957 | 0.59458 | 0.66942 | 0.72418 |

Table 7: Contact prediction performance of the full SSA model on the SCOP ASTRAL 2.06 test set and the 2.07 new test set. We report precision, recall, F1 score, and the area under the precision-recall curve (AUPR) for predicting all contacts ($|i - j| \geq 2$) and distant contacts ($|i - j| \geq 12$) in the test set proteins. We also report precision of the top $L$, $L/2$, and $L/5$ predicted contacts.

In order to facilitate some comparison with state-of-the-art co-evolution based contact prediction methods, we also report results for contact prediction using our full SSA model on the publicly released set of free modelling targets from CASP12 [40]. This dataset consists of 21 protein domains. We include the complete list at the end of this section. We compare with deep convolutional neural network models using co-evolution features, RaptorX-Contact [41], iFold & Deepfold-Contact [42], and MetaPSICOV [43], and with GREMLIN [44], an entirely co-evolution based approach. We find

that our model dramatically outperforms these methods when predicting all contacts but performs worse when predicting only distant contacts (Appendix Table 8). This is unsurprising, as our model is trained to predict all contacts, of which local contacts are much more abundant than distant contacts, whereas the co-evolution methods are designed to predict distant contacts. Furthermore, we wish to emphasize that our model is tuned for structural similarity prediction and that our contact prediction module is extremely simple, being only a single fully connected layer followed by a single convolutional layer. It is possible that much better performance could be achieved using our embeddings with a more sophisticated contact prediction architecture. That said, our model does outperform the pure co-evolution method, GREMLIN, based on AUPR for predicting distant contacts. These results suggest that our embeddings may be useful as features, in combination with co-evolution based features, for improving dedicated contact prediction models on both local and distant contacts.

| Contacts | Model | Precision | Recall | F1 | AUPR | Pr @ L | Pr @ L/2 | Pr @ L/5 |
|---|---|---|---|---|---|---|---|---|
| All | GREMLIN | 0.75996 | 0.04805 | 0.07573 | 0.09319 | 0.23348 | 0.30969 | 0.36271 |
| | iFold_1 | 0.11525 | 0.3443 | 0.16631 | 0.20027 | 0.40381 | 0.48265 | 0.53176 |
| | Deepfold-Contact | 0.15949 | 0.30481 | 0.16092 | 0.19487 | 0.39047 | 0.44670 | 0.47191 |
| | MetaPSICOV | 0.51715 | 0.08935 | 0.13673 | 0.19329 | 0.42288 | 0.50202 | 0.58202 |
| | RaptorX-Contact | 0.41937 | 0.14135 | 0.19373 | 0.20501 | 0.43254 | 0.50384 | 0.56178 |
| | SSA (full) | 0.78267 | 0.47298 | 0.58091 | 0.61299 | 0.98773 | 0.99147 | 0.9928 |
| Distant | GREMLIN | 0.73652 | 0.06184 | 0.08973 | 0.07231 | 0.15211 | 0.22508 | 0.27663 |
| | iFold_1 | 0.10405 | 0.60654 | 0.17084 | 0.26605 | 0.33625 | 0.40596 | 0.46482 |
| | Deepfold-Contact | 0.14852 | 0.52808 | 0.16548 | 0.26589 | 0.3422 | 0.39832 | 0.41885 |
| | MetaPSICOV | 0.59992 | 0.13366 | 0.18350 | 0.25332 | 0.35483 | 0.43886 | 0.51106 |
| | RaptorX-Contact | 0.38873 | 0.23186 | 0.26050 | 0.29023 | 0.37874 | 0.45729 | 0.51607 |
| | SSA (full) | 0.35222 | 0.03950 | 0.06216 | 0.10226 | 0.16548 | 0.19138 | 0.22780 |

Table 8: Contact prediction performance of the full SSA model on the CASP12 free modelling targets with results of state-of-the-art co-evolution based methods for comparison. We report precision, recall, F1 score, and the area under the precision-recall curve (AUPR) for predicting all contacts ($|i - j| \geq 2$) and distant contacts ($|i - j| \geq 12$). We also report precision of the top $L$, $L/2$, and $L/5$ predicted contacts.

**CASP12 free modeling domains:** T0859-D1, T0941-D1, T0896-D3, T0900-D1, T0897-D1, T0898-D1, T0862-D1, T0904-D1, T0863-D2, T0912-D3, T0897-D2, T0863-D1, T0870-D1, T0864-D1, T0886-D2, T0892-D2, T0866-D1, T0918-D1, T0918-D2, T0918-D3, T0866-D1

