# OpenReview forum: "Learning protein sequence embeddings using information from structure"
_ICLR.cc/2019/Conference_

### Official Review · AnonReviewer1 · 2018-10-26
**Good application paper but evaluation must be strengthened**

**Rating:** 8
**Confidence:** 4

**Review:**

General comment
==============
The authors describe two loss functions for learning embeddings of protein amino acids based on i) predicting the global structural similarity of two proteins, and ii) predicting amino acid contacts within proteins. As far as I know, these loss functions are novel and the authors show clear improvements when using the learned embeddings in downstream tasks. The paper is well motivated and mostly clearly written. However, the evaluation must be strengthened and some aspects of it clarified. Provided that the authors address my comments below, I think it is a good ICLR application paper.

Major comments
=============
1. The authors should describe how they optimized hyperparameters such as the learning, lambda (loss section 3.3), or the smoothing factor (section 3.4). These should be optimized on an evaluation set, but the authors only mentioned that they split the dataset into training and holdout (test) set (section 4.1).

2. The way the authors present results in table 1 and table 2 is unclear. Both table 1 and table 2 contain results of the structural similarity tasks but with different baselines. ‘SSA w/ contact predictions’ is also undefined and can be interpreted as ‘with’ or ‘without’ contacts predictions. I therefore strongly recommend to show structural similarity results in table 1 and secondary structure results in table 2 and include in both tables i) ‘SSA full’, ‘SSA without contact predictions’, and ‘SSA without language model’.

3. The authors should compare SSA to the current state-of-the art in structure prediction in addition to baseline models.

4. The authors should evaluate how well their method predicts amino acid contact maps.

5. The authors should describe how they were dealing with variable-length protein sequences. Are sequences truncated and embedded to a fixed length? What is the mean and variance in protein sequence lengths in the considered datasets? The authors should point out that their method is limited to fixed length sequences.

6. The authors should briefly describe the training and inference time on a single GPU and CPU. How much memory is required for training with a certain sequence length, e.g. 400 amino acids per sequence? Does the model fit on a single GPU?

7. The authors should discuss limitations of their method, e.g. that it cannot handle variable length sequences and that the memory scales quadratically by the the sequence length.

8. CRF (SSA) (table 3) includes a biLSTM layer between SSA and the CRF. However, the biLSTM can learn a non-linear projection of embeddings learned by SSA such that it is unclear if improvements are due to the embeddings learned by SSA or the biLSTM+CRF architecture. The authors should therefore train a biLSTM+CRF model on one-hot encoded amino-acids and include it as baseline in table 3.


Minor comments
=============
9. The way the similarity score s’ is computed (section 3.2.1) should be motivated more clearly. Why do the authors compute the score s’ manually instead of predicting it, e.g. using a model that takes the embeddings z of both proteins as input and predicts a single scalar s’?

10. How does ordinal regression (section 3.2.2) perform compared with a softmax layer? Why do the authors compute s’ and then train logistic regression classifiers on s’ to predict the similarity level, instead of predicting the similarity level directly based on the embeddings z?

11. Why do the authors use a distance threshold of 8A (section 3.3)? Is this common practice in the field?

12. Why do the authors use the not product and the absolute difference as features instead of the embeddings z directly? Which activation function is used to predict contact probabilities (sigmoid, softmax, …)?

13. The authors should reference and describe the results presented in table 1 more clearly.

14. Optional: the authors should analyze if learned embeddings are correlated with amino acid and structural properties such as their size, charge, or solvent accessibility. Do embeddings clusters by certain properties? This can be analyzed, e.g., using a tSNE plot.

15. How does TMalign perform when using the maximum or geometric average instead of the arithmetic average of the two scores (section 4.1)

---

> ### Author Response · Authors · 2018-11-25
> **Response to the reviewer**
>
> We thank the reviewer for their detailed critique and helpful suggestions. We apologize for the lack of clarity in various aspects of the manuscript and have made revisions to address these concerns. We have also now included results for the requested additional experiments. Please see below for our responses to the reviewer’s specific comments.
>
> Major comments:
> 1. We have updated the manuscript to include an explanation (Section 3.4) of how hyperparameters were chosen. We note that performance on a validation set held-out from the training set and separate from the test set was used to select amino acid resampling probability and the loss interpolation parameter, lambda. Amino acid resampling probability was chosen to be 0.05 as we had observed a small improvement in performance on the validation set for models trained with resampling over models trained without resampling. Lambda was chosen from {0.5, 0.33, 0.1} to give best structural similarity prediction accuracy on the validation set. We observe that decreasing lambda corresponded to increasing accuracy over this range. (See Appendix A.2 for more details)
>
> We wish to emphasize that the rest of the hyperparameters were not tuned based on performance on any data, (held-out or otherwise), and we would expect that the results should be insensitive to reasonable settings, In particular, learning rate was set to 0.001 following common practice with the ADAM optimizer and we observed good convergence on the training set with this setting. The smoothing factor was set to 0.5 in order to slightly upweight pairs of sequences with high similarity that would otherwise be rare during in trining. In particular, we choose 0.5 in order to have roughly two examples of family level similarity per minibatch.
>
> 2. Table 2 is an ablation study of the model components. Table 1 is a comparison with other protein similarity scoring methods. We now make these points clearer in their corresponding sections and table captions.
>
> 3. In order to use structure prediction for predicting SCOP co-membership given a pair of protein sequences, we would have to first predict structure for each sequence and then compare the predicted structures to predict co-membership using TMalign. This seems redundant, because we have already compared with TMalign using the actual structures for proteins in our datasets and found that our full SSA model is significantly better at predicting SCOP co-membership. That said, it would be interesting to investigate how predicted structures may perform. However, these methods require significant time per protein (I-TASSER reports 1-2 days per structure), so generating predictions for our ASTRAL 2.07 new test set of 688 sequences is not possible before the end of the review period, let alone predicting structures for all 5,602 sequences in our ASTRAL 2.06 test set.
>
> 4. Our focus was not to train the best contact map prediction method, but rather to use the contact prediction task to better embed local structure at each position and to improve the structure similarity prediction task. Because the model is predicting contacts directly from raw sequence, we think it is unlikely to perform as well as the best co-evolution based methods. Nonetheless, we have now included the contact prediction performance results in the Appendix.
>
> 5. Variable length sequences require no special consideration with our model. The biLSTM naturally handles sequences of arbitrary length. During training, no truncation of any kind is performed. Our model is not limited to fixed length sequences. We now make this clearer in the manuscript. In terms of sequences lengths, the mean length in the training set is 175.9 amino acids with a standard deviation of 110.1 amino acids. The minimum and maximum lengths are 20 and 1,449 respectively. The mean length in the 2.06 test set is 179.7 with a standard deviation of 114.3 and minimum and maximum sequence lengths of 21 and 1,500. In the 2.07 new test set, the mean length is 190.3 with a standard deviation of 148.7 and minimum and maximum lengths of 25 and 1,664.

---

> > ### Author Response · Authors · 2018-11-25
> > **Response continued**
> >
> >
> > 6. Embedding time is very fast (~0.03 ms per amino acid on an NVIDIA V100) and easily fits on a single GPU (requiring <9GB of RAM to embed 130,000 amino acids) with both time and memory scaling linearly with sequence length. Computing the SSA for sequence comparison scales as the product of the sequence length, O(nm), but is easily parallelized on a GPU. Computing the SSA for all 237,016 pairs of sequences in the 2.07 new test set required on average 0.43 ms per pair (101 seconds total)  when performed serially on a single NVIDIA V100 GPU. Training was performed on a single NVIDIA V100 and required ~3 days to train. We now include this description in the manuscript.
> >
> > 7. We are afraid there has been a misunderstanding which we hope to have now resolved in the manuscript. The model easily handles variable length sequences. This is not a limitation of our model. During training and prediction, the memory required to embed an amino acid sequence into a sequence of vectors scales linearly with the sequence length. The memory required to form the pairwise similarity matrix between two sequences scales as the product of their lengths (one entry is required for each pair of positions). Contact prediction does require memory scaling quadratically with sequence length (one entry for each pair of positions within the sequence). One real limitation is that the model is trained on protein domains and the encoder gives embeddings that are highly contextualized. This means that if the model is run on multidomain protein sequences it isn’t clear that the domains will map to the same embedding vectors as the domains in isolation. On the one hand this makes sense and is desirable, because we know structure is highly context dependent. On the other hand, we might think that the structure of domains should be the same regardless of surrounding context and that these sequences should receive the same embeddings as a result. There is definitely room for further work in this area.
> >
> > 8. We have updated the manuscript to include the requested experiment in table 3. To address the reviewer’s concern, the biLSTM+CRF with 1-hot encoded amino acids as features performs poorly with an overall score of 0.52, barely matching the performance of MEMSAT-SVM overall. This result indicates that the biLSTM+CRF architecture is not enough to achieve good performance.
> >
> >
> > Minor comments:
> > 9. s’ is defined by the learned embeddings and in this sense is learned. The reason we do not pass the embeddings of each sequence into a separate model to predict the similarity is two-fold. (1) we want the learned embeddings to have an interpretable correspondence – i.e. the distance between two embeddings encodes their semantic similarity. Feeding the embeddings directly into some unconstrained model would not impose this condition. Furthermore, we want all of the “comparative structural information” to be encoded directly into the embeddings rather than potentially emerging from the downstream model. (2) the sequences are of variable length so it isn’t clear how such a model should be structured, but, in section 4.2, we compare against two alternative methods for comparing between sequences of vector embeddings.
> >
> > 10. The motivation behind using ordinal regression is that it explicitly captures ordering information about the similarity labels (similarity 0 is less than similarity 1 is less than similarity 2 etc.). The ordinal regression framework imposes this property directly in the structure of the model and ensures that predicted similarity increases monotonically with the alignment score.

---

> > > ### Author Response · Authors · 2018-11-25
> > > **Response (final part)**
> > >
> > >
> > > 11. Distance thresholds anywhere in the range of 6-12a are common. We chose 8a somewhat arbitrarily from this range based on the average size of an amino acid.
> > >
> > > 12. In the contact prediction component, we use the elementwise products and absolute differences between the embeddings as features rather than the concatenated features primarily because we want the feature vectors to be symmetric (h_ij = h_ji). This featurization has shown success for semantic relatedness in NLP in papers like “Improved Semantic Representations From Tree-Structure Long Short-Term Memory Networks” Tai et al. 2015. We use sigmoid activation to give the predicted contact probabilities.
> > >
> > > 13. We have revised this section in order to improve the clarity of table 1.
> > >
> > > 14. This would be an interesting analysis to complement secondary structure prediction for examining whether the embeddings are capturing local structural properties of the sequence. However, space and time constraints make this difficult to include. We are definitely interested in following up on this direction in future work.
> > >
> > > 15. TMalign actually performs slightly worse when using the maximum or geometric mean instead of arithmetic mean. With geometric mean we get 0.80729 accuracy, 0.60197 Pearson’s correlation, and 0.37059 Spearman’s correlation on the 2.06 test set and 0.8126 accuracy, 0.80102 Pearson’s correlation, and 0.38548 Spearman’s correlation on the 2.07 test set. Using the maximum of the two scores performs even worse getting 0.79255 accuracy, 0.51852 Pearson’s correlation, and 0.28407 Spearman’s correlation on the 2.06 test set and 0.80742 accuracy, 0.75921 Pearson’s correlation, and 0.34023 Spearman’s correlation on the 2.07 test set.

---

> > > > ### Comment · AnonReviewer1 · 2018-11-28
> > > > **Manuscript clearly improved; Only few minor follow-up comments.**
> > > >
> > > > I appreciate that you rigorously addressed all my comments. I have only few minor follow-up comments. I will increase my rating once you addressed these outstanding comments.
> > > >
> > > > 4. Contact map prediction performance
> > > > Thank for including contact map prediction results in Appendix A5. Please reference Appendix A5 (and all other sections) in the main text. Can you also include state-of-the art results? You can consider showing performance number in a table for clarity. I agree that your model is not tweaked for contact map prediction but it is informative to know its performance relative to the state of the art for assessing if embeddings can be used as (additional) input of a contact map predictor.
> > > >
> > > > 7. Discussion of limitations
> > > > Thanks for clarifying that your model can handle variable length sequences, which is certainly an advantage. Can you briefly discuss the problem with multi-domain proteins in the conclusions sections?
> > > >
> > > > Page 8: The reference ‘Appendix Table ??’ is broken. Please fix it.
> > > >
> > > > I encourage you to publish your source code at the end of the review period.

---

> > > > > ### Author Response · Authors · 2018-12-04
> > > > > **Response to follow up**
> > > > >
> > > > > Unfortunately, we can no longer submit additional revisions to OpenReview. However, we have made the requested updates to the manuscript for the camera-ready version (assuming it is accepted!). We detail those updates and respond to the reviewer’s additional comments below.
> > > > >
> > > > > 4. Contact map prediction performance
> > > > > We have included a reference to Appendix A.5 in the manuscript and have updated those results to a table for easier examination. While we appreciate the reviewer’s interest in seeing state-of-the-art contact prediction performance and we too are interested to see if our embeddings could be used to improve contact prediction models, we want to emphasize that the primary goal of our model is structural comparison, and contact prediction is merely an auxiliary task. Furthermore, due to the nature of our train/test splits, it would be extremely difficult to correctly compare our contact prediction performance with the state-of-the-art methods on those datasets.
> > > > >
> > > > > That said, to give some idea of the differences in performance, we now include results for contact prediction using our full SSA model on the publicly released protein structures from CASP12 that were used for contact prediction. This is, unfortunately, a very small dataset, but it gives some idea of how our model compares with the best co-evolution based deep convolutional neural network models. We find that our model performs much better than these methods when predicting all contacts but performs worse when predicting only distant contacts (|i-j| > 11). This is, perhaps, expected, because the co-evolution methods focus on distant contact prediction, whereas our model is trained to predict all contacts, of which the vast majority are local. Interestingly, we do achieve similar distant contact prediction performance to GREMLIN, a purely co-evolution based model (no neural network). To us, this suggests that our embeddings are likely to have utility as additional inputs to the state-of-the-art models. We have included these results and discussion in Appendix A.5.
> > > > >
> > > > > 7. Discussion of limitations:
> > > > > We have added some discussion of single- vs. multi-domain proteins in the conclusion.
> > > > >
> > > > > The broken table reference has also been fixed. Thank you for pointing this out.
> > > > >
> > > > > We plan to release source code with the camera-ready version of the manuscript!

---

### Official Review · AnonReviewer3 · 2018-11-03
**Promising idea but falls short in write-up / evaluation**

**Rating:** 7
**Confidence:** 3

**Review:**

Thanks for the detailed responses. After reading the author response and the updated paper, I am satisfied on several of my concerns, many of which were due to the writing in the earlier submission. The updated results on various comparisons are also good.  I have updated my score accordingly. Some qualitative analysis of the results would have been nice -- examples of protein pairs where they do well and other methods have difficulty as they don't use the structural similarity info / global sequence info used by this paper. But maybe those can be in a journal submission.
My only remaining concern is on the lack of reporting average performance on the test data (which used to be the norm until recently for papers submitted to ML conferences).


Summary:
This paper proposes an approach for embedding proteins using their amino-acid sequences, with the goal that embeddings of proteins with similar structure are encouraged to be close in the embedded space. A stacked 3-layer Bi-directional LSTM is used for embedding the proteins. The structure information is obtained from the SCOP database which is used in an ordinal regression framework, where the output is the structural similarity and inputs are the embeddings. Along with the ordinal regression, another loss term to incorporate contacts of amino-acid residues is used. Results are shown on structural similarity prediction and secondary structure prediction.

Clarity:
1. The introduction of the paper is not very well written and it takes some time to figure out the exact problem being addressed. Is it learning sequence embeddings, or predicting structure from sequence or searching for similar structures in a database. Defining a clear goal -- input/output of their pipeline is important before describing the applications of the method, such as predicting structural similarity.
2. Due to the write-up, the method comes across as having too many modeling components without a very clear motivation for why these help the problem at hand. Where is the alignment part?
3. Why is each sequence embedded as a matrix? What is the motivation for a vector representation at each amino-acid position?
4. The authors need to explain the particular choice of 3 layers of bi-directional LSTMs. Why three? And why Bi-LSTM and not LSTMs?

Quality:
1. While the problem being addressed is interesting, the work lacks a clear reasoning behind the choice of modeling components which makes it seem ad-hoc.
2. Structural similarity is defined using the hierarchy of protein structure classes and the numbers seemed a bit arbitrary to me. Why not have a vector to encode the different aspects of structure? Have they looked at prior work?
3. How does the pre-trained language model on Pfam sequences help? Why is the output from it concatenated; have other composition functions been considered?

Originality:
The various components of the model are not novel, but the particular framework of putting them together is novel.

Results:
1. While the authors claim that some prior methods only work with high sequence similarity, their own evaluation only considers pairs of sequences with 95% identity. HHalign for instance, considers sequences with ~20% identity.
2. Why weren't several train/test splits of the data tried, so that performance can be reported with std. error bars?
3. Methods against which they compare have not been described properly.

---

> ### Author Response · Authors · 2018-11-25
> **Response to the reviewer**
>
> We apologize to the reviewer for the lack of clarity in the manuscript particularly regarding the formulation of the problem. We have made significant revisions to the manuscript to improve clarity and better justify the specific modeling decisions. Other than presentation, we feel that many of the reviewer’s concerns are already addressed in the manuscript but perhaps were not made clear in our writing. We hope that our revised manuscript improves on this. We address the individual concerns of the reviewer below.
>
> Clarity:
> 1. The goal is to learn sequence embeddings using structural information. A secondary goal is to have a model that is good at predicting structural similarity from sequence. Structure prediction is not a goal – in fact, we are specifically trying to avoid doing structure prediction. We have updated our manuscript in an effort to improve our description of the problem.
>
> 2. The alignment part is described in section 3.2.1 of the manuscript and also appears in Figure 1. It is used to define a scalar similarity between two amino acid sequences using their sequences of vector embeddings.
>
> 3. The motivation for learning a vector representation of each amino acid position is to try to capture local structure information (based on sequence). Primarily, this allows the embeddings to be used as features in sequence to sequence prediction problems (like transmembrane prediction) that would not be possible with single vector representations of the sequence. It also gives interpretability in the sense that we can ask how sections of sequence or individual positions differ between protein sequences.
>
> 4. The particular architecture of a 3-layer biLSTM allows embeddings to potentially be functions of distant sequence elements. Appendix Table 4 shows a comparison of a few embedding architectures. Specifically, we compare linear, fully connected, and 1-layer biLSTM models. We have clarified this detail in section 4.2. Single direction LSTMs would not be able to capture whole sequence information when forming the embedding at each position but only information on one side of that position (i.e. z_i would be a function of amino acids up to position i but not after position i if a single direction LSTM encoder was used). Clearly, local structure depends on the sequence to either side.
>
> Quality:
> 1. We have updated the manuscript to better justify the specific modeling components.
>
> 2. It’s unclear what exactly the reviewer is suggesting here. The goal is to learn embeddings that capture structural information as a function of sequence. At “prediction time” only sequences are observed. If the question is: why not try to predict specific structural properties, again, it’s not clear what properties the reviewer has in mind. We are trying to predict the properties that give rise to the SCOP classification. The SCOP category membership represents expert knowledge in protein structure and already groups proteins by structural properties.
>
> 3. Although we don’t understand exactly why the pretrained language model helps, the intuition is that the language model hidden states capture information about the space of natural protein sequences that is useful for learning about structural similarity. It is transferring information about what natural proteins “look like” from a large set of proteins to this problem where we have a relatively smaller number of sequences. We show empirically in table 2 that the language model improves structure similarity prediction results.
>
> Originality:
> The language model is novel in application to protein sequences. The SSA component is novel for defining similarity between sequences and learning embedding models from observed global similarity. Using contact prediction for learning sequence embeddings is also novel.
>
> Results
> 1. The pairs of sequences considered have no more than 95% sequence identity. In fact, the vast majority of pairs of sequences have much less than this. The average percent identity between sequence pairs is 13% in both the ASTRAL 2.06 test set and ASTRAL 2.07 new test set. Furthermore, we compare with all baselines on exactly the same sequence pairs, so the comparison is valid. We now include this information in the manuscript.
>
> 2. We did not consider several train/test splits to be necessary, because the test split is already large (much larger than commonly used in biology applications) with 20% of the sequences (5,602) being held out. The test set composed of proteins added in the 2.07 release of ASTRAL is admittedly smaller (688 sequences), but still comprises many more sequences than commonly used in the field.
>
> 3. We have made an effort to better describe the baselines in the revised manuscript.

---

### Official Review · AnonReviewer2 · 2018-11-04
**Are protein sequence emebeddings learned from structure?**

**Rating:** 7
**Confidence:** 4

**Review:**

This work learns embeddings for proteins. They use techniques from deep learning on natural langauge that are typically applied to scentences and words, and apply them correspondinly to proteines and amino acids. Thus they learn a vector representation using a bi-directional LSTM for amino acids by training the amino acid equivalent of a language model.

The authors then multitask 2 models using the embeddings that perform contact prediction (using an mlp and CNN) and structural class similarity model, which appear to perform very well.

Their SSA - soft symmetric alignment mechanism is neat and gives a single scalar value for a pair of proteins (by comparing their strings of emebedded amino acids by L1 distance), and it is descriptive enough feature for a simple ordinal regression to output a set structural similarity scores via a linear classifier (one for each strength of similarity re. the SCOP classification hierachy). It seems to work well, but I am unable to judge how good this is with respect to more recent work in this field. I would suspect being able to backprop to the embedding LSTMs through the SSA at this point would give much better results.

Authors only give 2 recent refeneces for protein embedding work [12,13] but should also take a look at this work: Melvin et. al, PLOS Computational Biology, this work uses structural class labels from SCOP to supervise the embedding. Although they do mention profile HMM in 'related work' which was used to create features in that work.

These authors, as far as I can tell do not "backprop" to the amino acid embeddings (and the LSTMs) from the contact or similarty loss. So the bi-LSTM-produced feature vectors, although trained unsupervised from many proteins, are not trained with structural supervision as claimed in the title (they state this in last paragraph of 3.1) and so the embeddings are not related to structural similarity directly. They do, however, seem to produce good features for the tasks they then tackle.
They say in the conclusion that the SSA model is fully differentiable, but I don't see where they "backprop" through it.

I would say (if this assessment is correct) then the title is very misleading, although the work and  final results look good.


update: the authors have assured me in comments that the model is trained end to end - changing rating to good..

---

> ### Author Response · Authors · 2018-11-25
> **The embedding model is learned from structure**
>
> We thank the reviewer for their comments and apologize for the confusing nature of some parts of the manuscript. We have revised the manuscript with the goal of improving clarity and hope that the new version will give a better understanding of our work. We address the reviewer’s specific comments below.
>
> 1. We are afraid there has been a misunderstanding about backprop through the LSTM embedding model. The model is trained end-to-end-- loss from the two tasks (structural similarity prediction and contact map prediction) are both back-propagated through their respective modules as well as through the LSTM embedding model. To emphasize, the LSTM embedding model is updated using strutural information, and the embeddings are in fact learned from structure. We note that in section 4.2, we show that a baseline model using only a linear transformation of the LM hidden states has far worse performance.
>
> 2. We have updated the manuscript to include missing related work, specifically “Detecting Remote Evolutionary Relationships among Proteins by Large-Scale Semantic Embedding” Melvin 2011. However, as pointed out by the reviewer, this work does fall generally into the category of methods using direct sequence alignment tools. The authors learn a low dimensional projection of a feature vector defined by  alignment of the query sequence against a database of protein domains using either psiblast or HHalign. Thus, their method gives single vector embeddings of protein domains based on existing sequence alignment tools in contrast to our work in which we learn a model that gives a vector representation for each position of the protein sequence using raw sequence as features.

---

### Meta-Review · Area_Chair1 · 2018-12-14
**Clear accept**

**Confidence:** 5
**Recommendation:** Accept (Poster)

**Metareview:**

The reviewers and authors had a productive conversation, leading to an improvement in the paper quality. The strengths of the paper highlighted by reviewers are a novel learning set-up and new loss functions that seem to help in the task of protein contact prediction and protein structural similarity prediction. The reviewers characterize the work as constituting an advance in an exciting application space, as well as containing a new configuration of methods to address the problem.

Overall, it is clear the paper should be accepted, based on reviewer comments, which unanimously agreed on the quality of the work.